# Dynamic control of hybrid grafted perfect vector vortex beams

Hammad Ahmed [1], Muhammad Afnan Ansari [1], Yan Li[1,2], Thomas Zentgraf [3], Muhammad Qasim Mehmood [4] & Xianzhong Chen [1] ✉

Perfect vector vortex beams (PVVBs) have attracted considerable interest due to their peculiar optical features. PVVBs are typically generated through the superposition of perfect vortex beams, which suffer from the limited number of topological charges (TCs). Furthermore, dynamic control of PVVBs is desirable and has not been reported. We propose and experimentally demonstrate hybrid grafted perfect vector vortex beams (GPVVBs) and their dynamic control. Hybrid GPVVBs are generated through the superposition of grafted perfect vortex beams with a multifunctional metasurface. The generated hybrid GPVVBs possess spatially variant rates of polarization change due to the involvement of more TCs. Each hybrid GPVVB includes different GPVVBs in the same beam, adding more design flexibility. Moreover, these beams are dynamically controlled with a rotating half waveplate. The generated dynamic GPVVBs may find applications in the fields where dynamic control is in high demand, including optical encryption, dense data communication, and multiple particle manipulation.

Perfect vector vortex beams (PVVBs) are a particular type of structured beams that possess peculiar properties of both vortex beams and vector beams[1]. The vortex component in a PVVB has a spiral phase wavefront that can carry orbital angular momentum (OAM), while the vector nature means that the beam has a spatially variant polarization profile[2]. PVVBs have been extensively used for particle manipulation[3], micro-drilling[4], and optical encryption[5,6]. PVVBs are typically generated through the superposition of orthogonal circularly polarized (CP) perfect vortex beams with different topological charges (TCs)[1,7–9]. However, this approach suffers from the limited number of topological charges involved since each vortex beam has only one TC, which is a fundamental challenge in the design. Furthermore, the dynamic control of these beams is highly desired to meet the requirement of time-varying systems.

Recently, grafted vortex beams (GVB) have attracted considerable attention due to the involvement of multiple TCs in the same beam, leading to distinct optical properties such as versatile OAM distributions[10]. The idea is inspired by the horticultural grafting in plants, i.e., the act of placing a part of one plant on a branch or root of a second plant. Similarly, a GVB can be formed by grafting two or more optical vortices (OVs) with different combinations of TCs. As a result, various combinations of TCs with different signs can be included in an individual GVB, which can overcome the limited number of TCs in the superposition of vortex beams and provide extra degrees of freedom in the design. By introducing an additional phase profile of a lens and that of an axicon, a grafted perfect vortex beam (GPVB) with a ring radius independent of its TC can be generated[11]. Current systems of generating and manipulating these beams rely heavily on complex optical setups due to the involvement of various bulky optical components, leading to high cost and large space requirement[12,13].

Benefiting from the unprecedented manipulation of phase, amplitude, and polarization[14–17], optical metasurfaces have provided a compact platform to realize various vortex beams[18–20] such as perfect

[1]Institute of Photonics and Quantum Sciences, School of Engineering and Physical Sciences, Heriot-Watt University, Edinburgh EH14 4AS, UK. [2]School of Materials, Zhengzhou University of Aeronautics, 450015 Zhengzhou, China. [3]Department of Physics, Paderborn University, Warburger Str. 100, 33098 Paderborn, Germany. [4]MicroNano Lab, Electrical Engineering Department, Information Technology University (ITU) of the Punjab, Ferozepur Road, Lahore 54600, Pakistan. ✉e-mail: x.chen@hw.ac.uk

vortex beams[11,21], ring vortex beams[22–24], OAM holography[25–27], vector beams[28–31], and vector vortex beam (VVBs)[28–31]. Recently, we have experimentally demonstrated a compact platform to generate and manipulate GPVBs[32]. However, these beams possess a homogeneous polarization state, meaning that 2D polarization control is completely inaccessible. A spatially variant polarization profile is expected in a grafted perfect vector vortex beam (GPVVB), which has not been reported due to its complex nature.

In this work, we propose to use a metasurface to generate a hybrid GPVVB, which can combine multiple GPVVBs on the same light beam. The hybrid GPVVB has the attributes of multiple GPVVBs based on the different combinations of TCs and polarization orders (polarization distribution per round trip). The generated beams can be dynamically tuned with a rotating half waveplate (HWP). A GVB possesses multiple TCs compared to that in a conventional vortex beam, overcoming the fundamental challenge in the design. More TCs are involved in the GVBs, providing extra degrees of freedom in the generation of hybrid GPVVBs. Moreover, our scheme offers a dynamic control over the generated hybrid GPVVBs with multiple polarization orders. This work will propel the state-of-the-art techniques for the VVB manipulation through the superposition of vortex beams and broaden its applications to many fields such as light–matter interaction, optical encryption, dense data communication, and particle manipulation.

## Results

Figure 1 shows the schematic of the proposed metasurface for the hybrid GPVVB generation. A linear polarization state can be decomposed into two different circular polarization states. When a linearly polarized light beam impinges onto the metasurface, a resultant hybrid GPVVB with a hybrid polarization distribution is produced through the superposition of hybrid GPVBs as shown in Fig. 1a, b. The vector nature of the hybrid GPVVB is evaluated with an analyzer (linear polarizer), whose transmission axis is fixed along the vertical direction as illustrated in the Fig. 1c. By continuously rotating the fast axis of the HWP, the polarization distribution of the generated beam is dynamically controlled (Fig. 1d). Different rotations of the lobes in the same intensity profile indirectly confirm the corresponding polarization property. Interestingly, the number of lobes in each sector of the GPVVB can be separately engineered with different combinations of TCs.

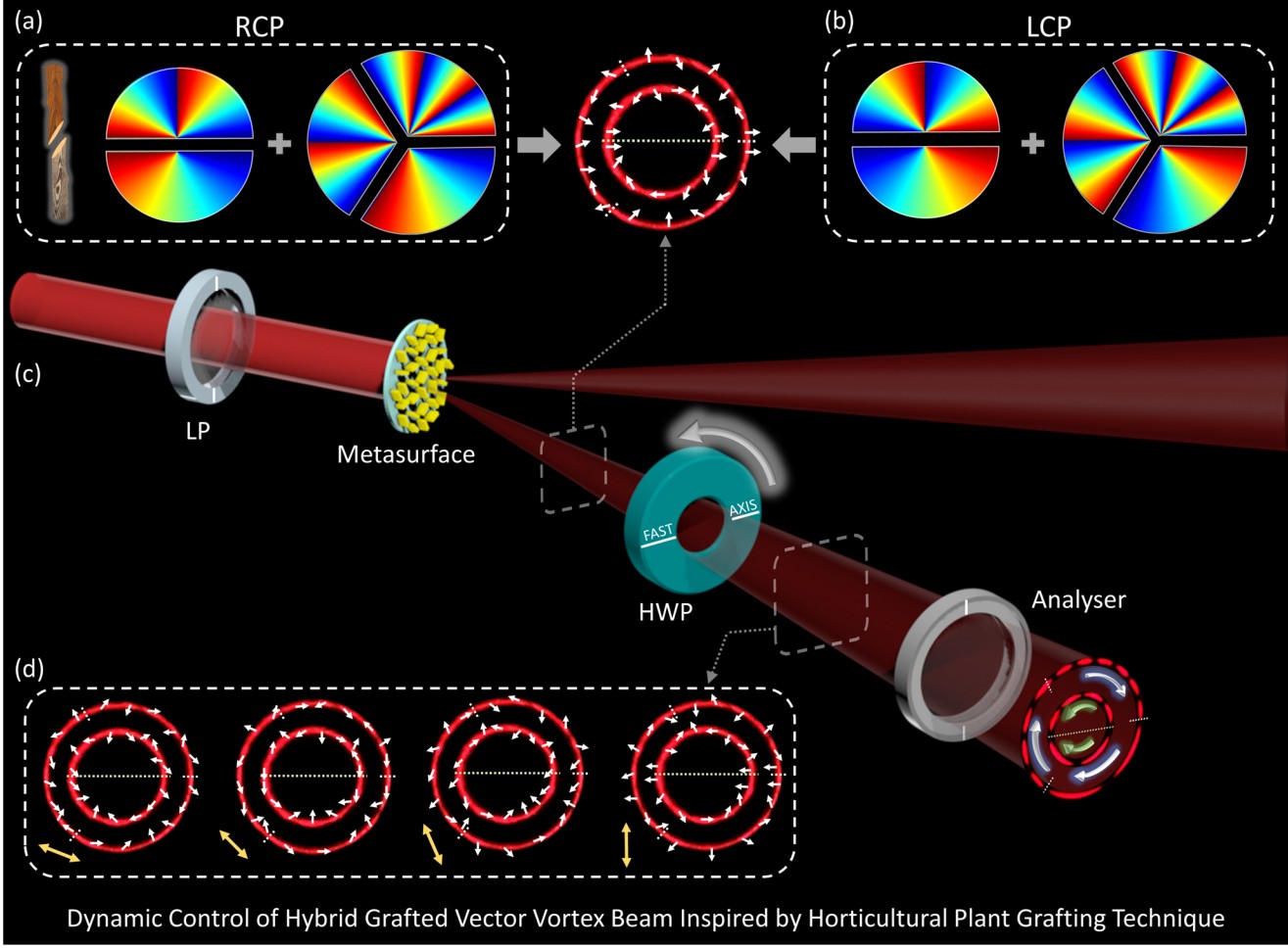

**Fig. 1 | Illustration of the approach for the dynamic control of hybrid GPVVBs. a, b** Inspired by the concept of horticultural plant grafting, the act of placing a part of one plant on a branch or root of another (left inset of (**a**)), GPVBs can be generated by grafting two or more phase profiles of GVBs on the same beam. The proposed hybrid GPVVBs here include two phase profiles of GPVBs with different concentric rings and combinations of TCs. The gaps in the phase profiles clearly show the different combinations of TCs for RCP (**a**) and LCP (**b**). **c** Upon the illumination of linearly polarized light at normal incidence, a hybrid GPVVB is generated through the superposition of two hybrid GPVBs with different circular polarization states. The white arrows represent the generated polarization profile in each sector of the hybrid GPVVB. A HWP with a rotating fast axis is added to dynamically modify the generated polarization profile. **d** Various polarization distributions of hybrid GPVVB are generated by changing the fast axis (indicated by yellow arrows) of the HWP. However, these beams have the same intensity profiles (two light rings). The polarization distributions are confirmed with a polarizer (analyzer), which will modify the intensity profiles with unique features (different distribution of lobes). Different rotation directions of lobes in the same intensity profile are clearly observed in (**c**).

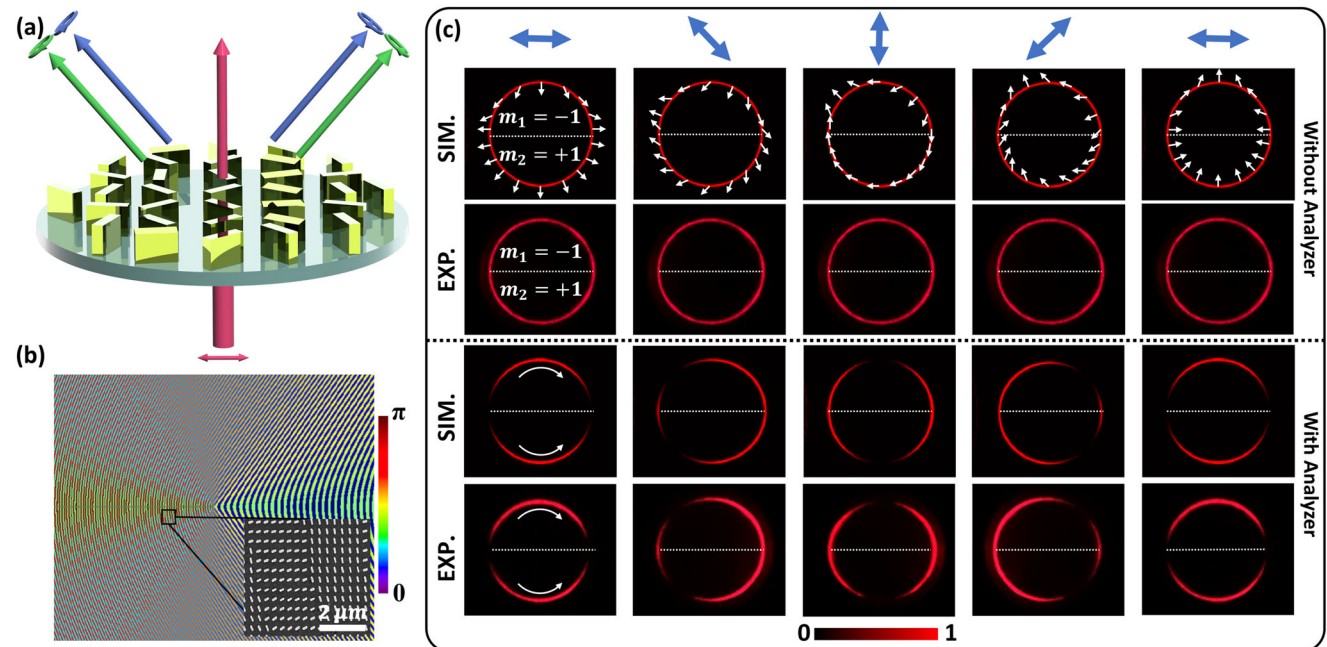

**Fig. 2 | Mechanism and device characterization. a** Upon the illumination of linearly polarized light, two symmetrically distributed GPVVBs are generated through the superposition of GPVBs with different circular polarization states. The metasurface consists of gold nanorods with spatially variant orientation angles on a glass substrate. **b** Orientation angle distribution of the metasurface with an area of $200 \times 200\ \mu m^2$. The inset shows the SEM image of a portion of the fabricated metasurface. **c** simulated and measured intensity profiles for the polarization order $m_1 = -1$ (upper sector) and $m_2 = +1$ (lower sector). The white arrows depict various polarization profiles under the illumination of linearly polarized light with different polarization directions (blue arrows). 1st and 2nd rows are intensity profiles without an analyzer, which are modulated with an analyzer (shown in 3rd and 4th rows). The transmission axis of analyzer is fixed along the vertical direction. Curved arrows show the direction of rotation of the lobes.

To realize such a beam, we start from the GPVVB generation, which can be represented as the superposition of two GPVBs with different circular polarization states (See Supplementary Section 1).

$$\boldsymbol{E_{GPVVB}} = E_R e^{i(\varphi_o + \alpha)} |\boldsymbol{R}_{GPVB_a}\rangle + E_L e^{-i(\varphi_o + \alpha)} |\boldsymbol{L}_{GPVB_b}\rangle, \quad (1)$$

where

$$|\boldsymbol{R}_{GPVB_a}\rangle = \exp\left[-\frac{(\rho - \rho_o)^2}{\Delta\rho^2}\right] e^{i\varphi_{GVB_a}} \begin{bmatrix} 1 \\ -i \end{bmatrix},$$

and

$$|\boldsymbol{L}_{GPVB_b}\rangle = \exp\left[-\frac{(\rho - \rho_o)^2}{\Delta\rho^2}\right] e^{-i\varphi_{GVB_b}} \begin{bmatrix} 1 \\ i \end{bmatrix}$$

are the GPVBs with right circular polarization (RCP) and left circular polarization (LCP), respectively. $\rho_o$ and $\Delta\rho$ are the radius and width of the ring, respectively. $\varphi_o$ is the initial phase and $\alpha$ is the angle between the transmission axis of the input linear polarizer (LP) and the x-axis (See Supplementary Section 2). $E_R$ and $E_L$ are the amplitudes of RCP and LCP components, respectively. $\varphi_{GVB}$ can be given as

$$\varphi_{GVB} = \arg\left\{\exp\left[i\sum_{n=1}^{N} rect\left(\frac{N\psi}{2\pi} - \frac{N+1}{2} + n\right) l_n\psi\right]\right\}, \quad (2)$$

Here, $N$, $l_n$, and $\psi$ are the total number of GVBs, TCs and the azimuthal angle, respectively. The polarization distribution for a particular sector can be obtained by the polarization order $m_n = \frac{(l_{a_n} - l_{b_n})}{2}$ and the phase distribution is determined by the topological Pancharatnam charge $p_n = \frac{(l_{a_n} + l_{b_n})}{2}$. $l_{a_n}$ and $l_{b_n}$ are the designed TCs for $GVB_a$ and $GVB_b$, respectively. The schematic of our design is illustrated in Fig. 2a. To realize such beams, we use a plasmonic metasurface with an off-axis

design, which can circumvent the non-converted part of the light on the transmission side. The metasurface consists of gold nanorods with spatially variant orientation angles $\phi$, which are governed by

$$\phi = \frac{1}{2}\arg\left(E_o e^{i(\varphi_{GVB_a} + \delta + \varphi_{ax} + \varphi_o)} + E_o e^{i(\varphi_{GVB_b} - \delta - \varphi_{ax} + \varphi_o)}\right), \quad (3)$$

where $\delta$ is the phase difference between neighboring pixels that can achieve a phase gradient along the x-direction, which can deflect the generated GPVVB. $\varphi_{ax} = -\frac{\pi\sqrt{x^2 + y^2}}{u}$ is the phase profile for the axicon, where $u$ is the axicon period to control the radius of the generated intensity ring. Upon the illumination of a linearly polarized light beam, the resultant GPVVB is formed through the superposition of different CP states since a linear polarization state can be decomposed into RCP and LCP with the same amplitudes. Figure 2b shows the desired nanorod orientation profile of the metasurface to generate GPVVB by grafting two vortex beams with $l_{a_1} = -1, l_{a_2} = +1, l_{b_1} = +1$, and $l_{b_2} = -1$. Moreover, the values of the other parameters are $N = 2$, $\varphi_o = 0, \delta = \frac{\pi}{3}$ and $u = 4\ \mu m$. The inset of Fig. 2b shows an SEM image of the fabricated metasurface sample, which consists of gold (Au) nanorods with different rotation angles $\phi$ on a glass (SiO$_2$) substrate. The geometry of nanorods is based on our previous work[33]. Each nanorod is 0.08 μm wide, 0.2 μm long, and 0.04 μm thick. The sample has an area of $200 \times 200\ \mu m^2$ with a pixel size of $0.3 \times 0.3\ \mu m^2$. Detailed information about the unit cell design is provided in Supplementary Section 3. The nanofabrication details are given in the Experimental Details section. The details about the efficiency of metasurfaces are provided in Supplementary Section 4.

A supercontinuum laser source (NKT-SuperK EXTREME) is utilized to characterize the sample. The wavelength of the incident light is 633 nm in this work. First, the linearly polarized light with different polarization directions is generated by rotating the transmission axis

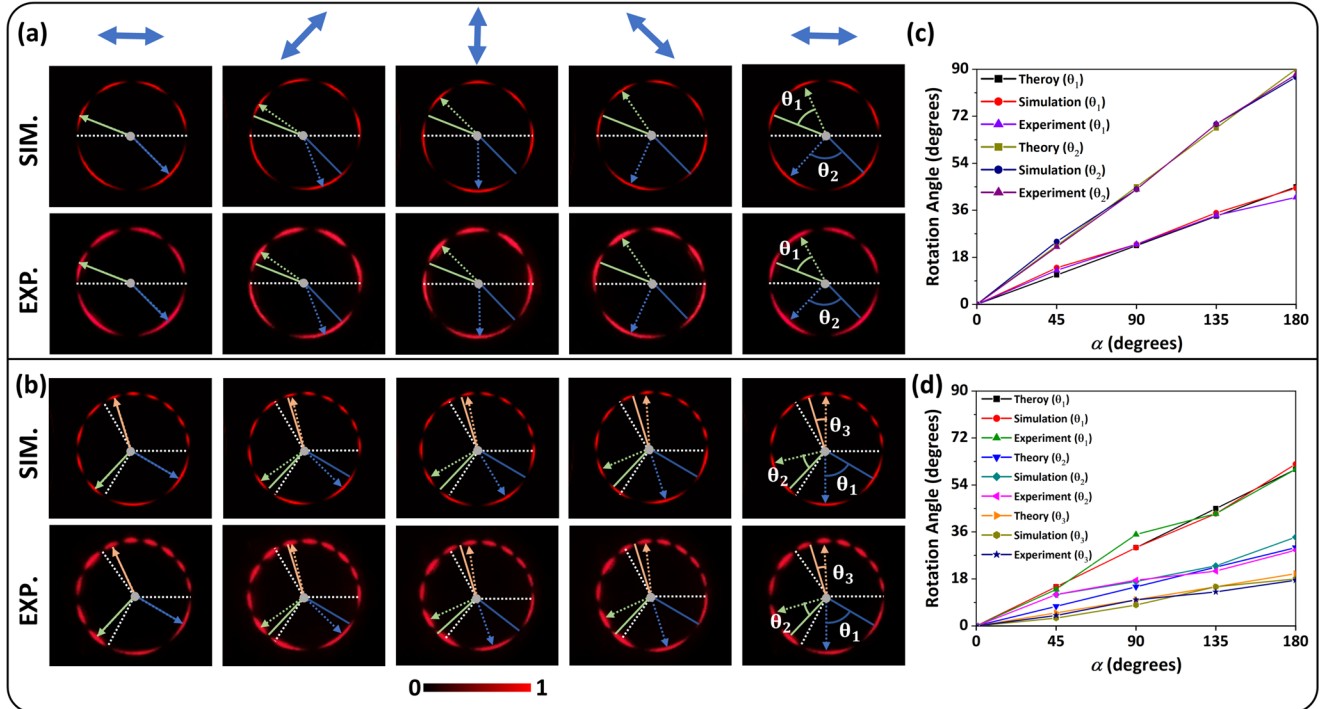

**Fig. 3 | Rotation angle measurement.** Simulated (top row) and experimental (bottom row) intensity patterns of GPVVBs after passing through an analyzer with a rotating transmission axis. GPVVBS are generated by grafting **a** two and **b** three vortex beams. The position of lobes in each sector can be rotated with different angles by controlling the transmission axis of the polarizer. $\theta_1$, $\theta_2$ and $\theta_3$ correspond to the rotation angles for each sector. White dashed lines indicate the boundary of a particular sector. The original positions of lobes are marked with blue, green, and orange solid lines, while the new positions of lobes are denoted with dashed lines. **c, d** show the relationship between rotation angles and α.

of an LP. A smartphone (Apple iPhone XS) is used to capture the intensity profile of the generated GPVVB on the observation plane. The detailed experimental setup is provided in Supplementary Section 5. The simulation of designed metasurfaces is performed based on the Kirchhoff diffraction integration[11], the details can be found in Supplementary Information Section 6. Based on the TCs chosen in Eq. (3), the generated GPVVB has $m_1 = -1$ (anti-vortex radial polarization), $p_1 = 0$ for the upper sector and $m_2 = +1$ (radial polarization), $p_2 = 0$ for the lower sector. Figure 2c shows the simulated (1st row) and experimentally (2nd row) measured intensity profiles with polarization distributions for different linear polarization states (0°, 45°, 90°, 135° and 180°). The unequal intensities in the experimental data without polarizer are mainly due to the imperfection of the optical setup (e.g., inaccurate alignment) and sample quality (e.g., missing nanorods). Blue arrows represent the polarization state of the incident light, whereas white arrows show the polarization distribution of the generated GPVVB. The analyzer with a transmission axis fixed along the vertical direction is used to evaluate the polarization distribution. The transmitted intensity patterns for a particular sector, after passing through the analyzer, are proportional to $\exp\left[-\frac{(\rho-\rho_o)^2}{\Delta\rho^2}\right]\cos^2\left(m_n\psi_n + \varphi_o + \alpha + \alpha_A\right)$. Here, $\alpha_A$ is the angle between the transmission axis of the analyzer and the x-axis. The 3rd (simulation) and 4th (experimentally measured) rows show the intensity patterns (with asymmetric distribution of lobes) revealed after the resultant beam passes through an analyzer. By continuously varying the incident polarization, the evolution of intensity patterns can be observed (see Supplementary Movie 1). The total number of lobes in a particular sector can be calculated by the expression: $\frac{|l_{a_n} - l_{b_n}|}{N}$. In this case, there are 2 lobes (one in each sector). Note that the rotation direction of a lobe is determined by the sign of the polarization order. To demonstrate the capability of complete dynamic control, both lobes are designed to have different rotation directions in

the same intensity ring. By gradually varying the incident linear polarization direction from 0° to 90°, the upper sector changes from an anti-vortex radially polarized beam to an anti-vortex azimuthally polarized beam, while the lower sector changes from a radially polarized beam to an azimuthally polarized beam. Similarly, when the incident polarization is varied from 90° to 180°, the anti-vortex azimuthally polarized beam (upper sector) and the azimuthally polarized beam (lower sector) return to the anti-vortex radially polarized beam and the radially polarized beam, respectively. As predicted in the simulation, the corresponding asymmetric rotation direction is clockwise for the upper sector and anti-clockwise for the lower sector, confirming the correct polarization distribution.

In the following, we explore the effect of incident polarization on the rotation speed of lobes of higher-order GPVVBs. For this purpose, two designs are considered. In the first design, a metasurface with polarization orders ($m_1 = +4, m_2 = +2$) is fabricated by grafting two OVs. While in the second design, another metasurface with polarization orders ($m_1 = +3, m_2 = +6, m_3 = +9$) is fabricated by grafting three OVs. The associated phase profiles and SEM images are provided in Supplementary Section 7. The rotation angles of the lobes in a particular sector of the GPVVB are theoretically calculated based on the relation

$$\theta_n = \frac{2\alpha}{Nm_n},\tag{4}$$

The above equation indicates that the rotation angles can be manipulated separately in each sector. Figure 3a, b show the simulated (top row) and experimental results (bottom row) of the GPVVB after passing through an analyzer (see Supplementary Section 8 and Supplementary Movies 2 and 3 for results without analyzer). The incident polarization (indicated by the blue arrows) is continuously altered by varying α from 0° to 180°(see Supplementary Movie 4). The colored

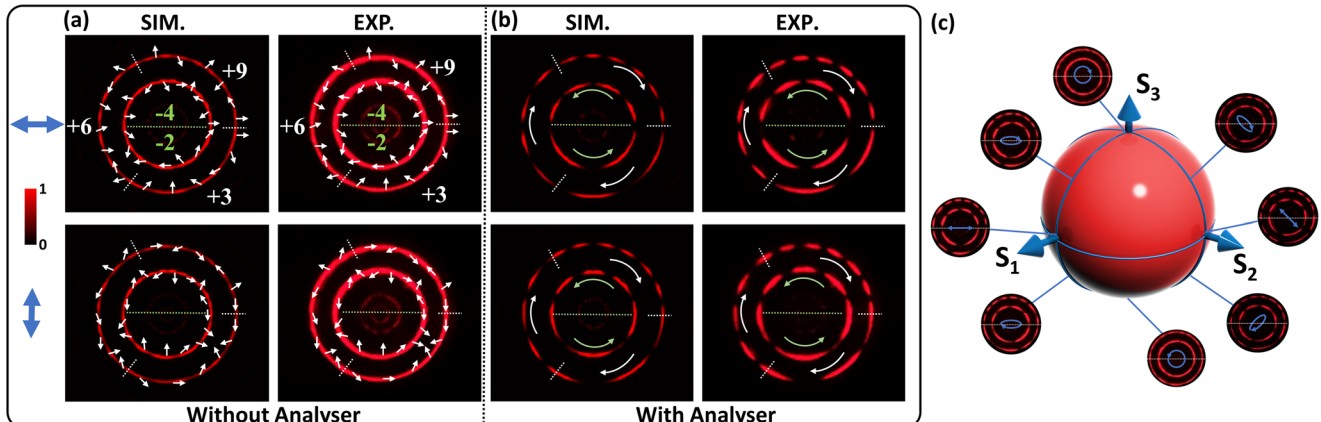

**Fig. 4 | Hybrid GPVVBs Generation. a** Intensity profile and polarization distribution of a hybrid GPVVB under the illumination of linearly polarized light along horizontal and vertical directions. Blue double arrows and white arrows represent the linear polarization direction of incident light and that on the hybrid GPVVB, respectively. **b** Modulated intensity profiles of the hybrid GPVVB after passing through an analyser. Rotation of lobes in different directions indicates the existence of various polarization orders. Rotation directions of inner and outer rings are shown in green and white curved arrows, respectively. **c** Effect of different polarization states on the hybrid GPVVB. A QWP is added after an input LP to generate various polarization states, which are mapped on the two different meridians on a HOPS. Blue arrows indicate the incident polarization states.

solid lines show the initial positions of lobes, which are located in the different sectors of the GPVVBs, while the colored dashed lines show the new positions of lobes. As $\alpha$ increases, the polarization profile of GPVVB changes, which results in the rotation of lobes in the clockwise direction. It is worth mentioning here that the rotation angles of the lobes in each sector are different. The difference can be seen with the increase of $\alpha$, which is further explained in Fig. 3c, d. When $\alpha$ is increased to 180° (in Fig. 3c), $\theta_1$ and $\theta_2$ are rotated by 45° and 90°, respectively. While in Fig. 3d, $\theta_1$, $\theta_2$ and $\theta_3$ are rotated by 60°, 30° and 20°, respectively. The slight discrepancy between theory and simulation is mainly due to the resolution of the observation plane in the Kirchhoff diffraction integration. The simulated rotation angles can approach theoretical values by increasing the resolution. The details of rotation angles against various $\alpha$ are provided in the Supplementary Section 9. Considering the chosen polarization orders, in the first design, the rotation speed in the lower sector is approximately twofold compared to that in the upper sector. On the other hand, in the second design, the rotation speed in the first sector is approximately threefold and twofold compared to that in the 2nd and 3rd sectors, respectively. Furthermore, the direction of rotation can be changed by changing the sign of $\alpha$ or $m_n$. In this way, the position of each lobe can be independently and precisely controlled by manipulating the incident polarization. This striking feature of the proposed GPVVBs can be potentially used to accurately position and accelerate the trapped particles in an optical tweezer.

Next, we experimentally demonstrate the generation of a hybrid GPVVB that is composed of two higher-order GPVVBs of different TCs and a combination of CPs. The corresponding expression can be written as

$$\phi = \frac{1}{2}\arg\Big\{E_o e^{i(\varphi_{GVB_a} + \delta + \varphi_{ax,u_1} + \varphi_o)} + E_o e^{i(\varphi_{GVB_b} - \delta - \varphi_{ax,u_1} + \varphi_o)} \\ + E_o e^{i(\varphi_{GVB_c} + \delta + \varphi_{ax,u_2} + \varphi_o)} + E_o e^{i(\varphi_{GVB_d} - \delta - \varphi_{ax,u_2} + \varphi_o)}\Big\}, \quad (5)$$

here $u_1 = 4\,\mu m$ and $u_2 = 5.5\,\mu m$ are the two axicon periods. This design is expected to generate the hybrid GPVVB with concentric rings of different polarization orders. The simulated and experimental intensity profiles along with their spatial polarization distributions under linear polarization incidence are elucidated in Fig. 4a. Blue double arrows and white arrows represent the linear polarization direction of incident light and that on the hybrid GPVVB, respectively. The spatial polarization profiles are calculated based on the Stokes polarimetry[1,15].

Details are provided in Supplementary Section 10. The same intensity profile contains multiple higher-order polarization distributions. The existence of small ring with very weak intensity is due to the parasitic light, which comes from the crosstalk between two rings[34–36]. The parasitic light can be minimized by optimizing the axicon period and the size of the metasurface. As an example, we simulate and experimentally realize a metasurface with an area of $400 \times 400\,\mu m^2$ to generate four higher-order GPVVBs. The small ring effect is dramatically suppressed. The simulation and experimental results are provided in Fig. S11 (Supplementary Section 11). Based on the given TCs ($l_{a_1} = -4, l_{a_2} = -2, l_{b_1} = +4, l_{b_2} = +2, l_{c_1} = +3, l_{c_2} = +6, l_{c_3} = +9, l_{d_1} = -3, l_{d_2} = -6,$ and $l_{d_3} = -9$), the inner ring has polarization order $m_1 = -4$ (upper sector), $m_2 = -2$ (lower sector), while the outer ring has $m_1 = +3$ (First sector), $m_2 = +6$ (2nd sector) and $m_3 = +9$ (3rd sector). Figure 4b shows the intensity pattern after passing through the analyzer. The number of lobes in different sectors of the intensity pattern is in good agreement with our theoretical prediction. The incident polarization is varied continuously to alter the polarization distribution in each sector (see Supplementary Movie 5). The resultant rotation directions for the inner and outer rings are anticlockwise and clockwise, respectively. Moreover, the effect of different polarization states on the hybrid GPVVB is depicted in Fig. 4c. A quarter waveplate (QWP) and the LP are used to generate the required polarization states. Eight polarization states, including RCP, 2 right elliptical polarizations (REP), horizontal linear polarization (HLP), 45°, 2 left elliptical polarization (LEP), and LCP are selected. The polarizations are geometrically illustrated by the points located along a meridian trajectory on the higher order Poincaré sphere (HOPS). The inset shows the measured intensity patterns at 633 nm. The CP states show a simple intensity profile containing two concentric rings. An elliptical polarization (EP) state, an intermediate state between CP and LP states, yields an intensity profile that is the combination of two polarization states. In this work, the intensity profile for EP contains lobes that are weakly connected.

To demonstrate the dynamic polarization control of the generated hybrid GPVVB, a HWP is added between the metasurface and the analyzer. The underlying mechanism is shown in Fig. 5a. The green arrow shows the original polarization direction. The angle between this direction and the fast axis (blue arrow) of HWP is $\beta$. The orange arrow depicts the new polarization direction, which is rotated by $2\beta$ in comparison with the original polarization direction. This offers another degree of freedom to dynamically control the polarization distribution of generated GPVVBs. With the help of Jones calculus, the

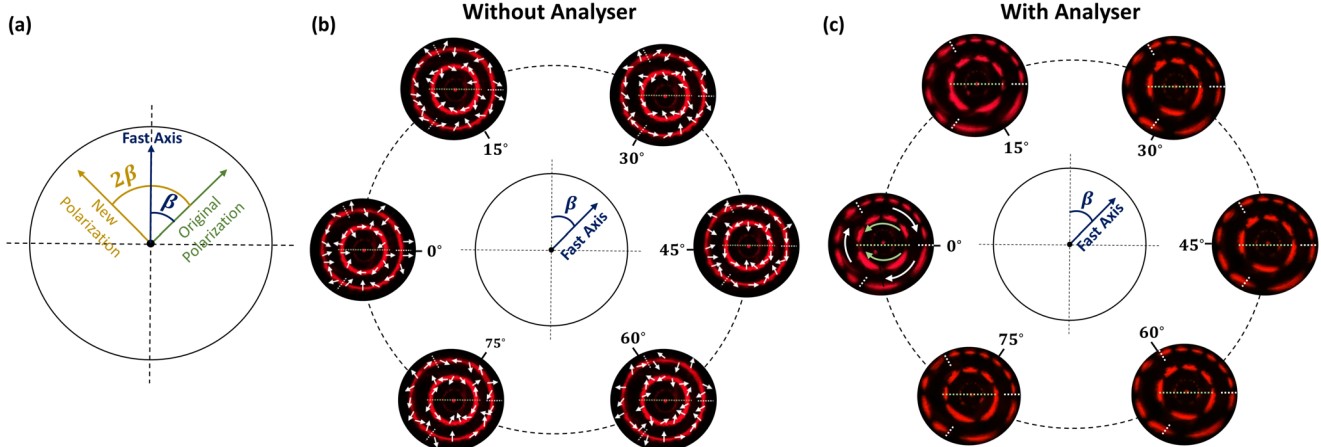

**Fig. 5 | Dynamic control of hybrid GPVVB Polarization. a** Polarization control mechanism. A linear polarization state with an inclined angle of $\beta$ (green arrow) with respect to the fast axis (blue arrow) of the HWP can be rotated to $2\beta$ (yellow arrow), which can be used to modulate the polarization profile of the hybrid GPVVB. **b** Measured intensity profiles and their modified polarization states by controlling the fast axis of the HWP. **c** Measured intensity profiles in (**b**) are modulated after the hybrid GPVVBs pass through the analyzer with a transmission axis along the horizontal direction.

modified GPVVB can be given as (see Supplementary Section 1 for detailed derivation)

$$E_{GPVVB_{new}} = J_{HWP}E_{GPVVB} = \exp\left[-\frac{(\rho - \rho_o)^2}{\Delta\rho^2}\right]e^{ip\psi}\begin{bmatrix}\cos(2\beta - m_n\psi - \varphi_o - \alpha) \\ \sin(2\beta - m_n\psi - \varphi_o - \alpha)\end{bmatrix}.$$

(6)

The above equation shows that the outgoing beam is still a GPVVB, but the polarization order is opposite to that of an incident beam and the polarization direction is rotated by $2\beta$. To experimentally verify this concept, a metasurface based on Eq. (2) is designed for a resultant beam with the polarization orders $m_1 = -4$, $m_2 = +2$ for the inner ring and $m_1 = +3$, $m_2 = +6$ and $m_3 = +9$ for the outer ring. Figure 5b shows the experimental intensity profiles along with polarization distribution at various $\beta$ (0°, 15°, 30°, 45°, 60°, 75°). The polarization distribution is confirmed when the light beam passes through an analyzer as illustrated in Fig. 5c. Based on the polarization orders, the outer ring is rotated clockwise with the increase of $\beta$, while the upper and lower sectors of the inner ring are rotated anticlockwise and clockwise, respectively.

## Discussion

We propose a generalized scheme to generate and manipulate a type of VVBs with unique features. In contrast to conventional VVBs/PVVBs, the proposed GPVVBs have the hybrid polarization information based on the different combinations of polarization orders and TCs. The intensity patterns of various GPVVBs are modulated (in the form of lobes) after the light passes through an analyzer. The unusual properties of the generated intensity patterns can be seen in Fig. 3, where the positions of the lobes in each sector of the light ring are precisely controlled by changing the incident polarization. The unique feature lies in the different rotation speeds of lobes in the same intensity profile, which is controlled by the polarization order. The direction of rotation can be manipulated by the direction of incident polarization and the sign of polarization order. Furthermore, the size of the lobes can also be modulated. The effect of $\alpha$ on the size of the lobe is analyzed in Fig. S12 (Supplementary Section 12). Two lobes, indicated by $P_1$, and $P_2$, are selected to study the change. The initial size of each corresponding lobe is denoted by $\gamma_1$ and $\gamma_2$. As the $\alpha$ increases from 0° to 180°, $P_1$ and $P_2$ start to rotate in opposite sectors, which causes an increase in $\gamma_1$ and a decrease

in $\gamma_2$. A linear relationship between $\gamma$ and $\alpha$ can be seen, which can be analytically given as $\gamma_1 = 32.404 + 2.4702\alpha$ and $\gamma_2 = 78 - 2.4579\alpha$. The corresponding correlation coefficients are 0.987 and 0.966, respectively. Hence, the results show that the size of the lobes can be linearly modulated by changing the incident polarization. In addition, our proposed approach provides the dynamic control over the generated hybrid polarization information, adding more degrees of freedom, which can further enhance the information capacity. The proposed hybrid GPVVB contain only two higher-order GPVVBs, which can be increased to incorporate more different combinations of TCs and polarization profiles. The maximum number of higher-order GPVVBs involved is mainly determined by the sample size and the design parameters of metasurfaces (e.g., pixel size). For instance, the increase in pixel number and the reduction of the pixel size of the metasurface can increase the number of GPVVBs. As an example, we simulate and experimentally realize a $400 \times 400$ $\mu m^2$ metasurface with four higher-order GPVVBs. The fabrication details and results (simulation and experimental) are provided in Figs. S7 and S11, respectively. The demonstrated metasurfaces consisting of gold nanorods have a low conversion efficiency, which can be dramatically increased with dielectric metasurfaces (e.g., titanium dioxide)[37]. As a proof of concept, the generated polarization profile of hybrid GPVVB is manipulated with an HWP. Liquid crystals and phase change materials-based design can be combined to build a more compact platform for sensing[38,39] and optical encryption applications[40,41]. This work offers not only an insight into structured light–matter interactions but also a degree of freedom to boost optical and quantum information capacity.

The uniqueness of the method and exotic properties of the resultant beams may find applications in many related research fields. For example, optical trapping can hold and rotate delicate biological samples, leading to the improvement in the 3D microscopic imaging of live cells by imaging samples from different directions[42,43]. However, existing techniques for holding and orienting cells have used complex optical apparatus[44] and have only been applied to single cells or small clusters of cells, limiting the practical applications within the community of cell biologists. Moreover, the proposed metasurface approach can trap multiple particles and provide more degrees of freedom to rotate the trapped biological structures. The availability of more TCs in a single beam and the dynamic control of VVBs can allow researchers in quantum science and optical communications to

encode more information in the light beams, which will dramatically increase the information capacity and offer more design flexibility.

In summary, we propose a metasurface approach to generating hybrid GPVVBs for the first time. The generated beams possess spatially variant rates of polarization direction change in the 2D space due to the more TCs involved. The hybrid GPVVBs are dynamically controlled with a rotating half waveplate, which can produce a tunable polarization profile based on a static metasurface without the need for complex switching. Our approach features the generation of hybrid GPVBs and the superposition of these beams with different circular polarization states for the realization of hybrid GPVVBs. We demonstrate a compact metasurface device to perform a fundamentally challenging task that is impossible with conventional optics. The engineered topological charges and polarization orders in the hybrid GPVVBs provide more degrees of freedom to tackle the fundamental challenge in the previous PVVB generation. This work opens an avenue for the generation of PVVBs and their dynamic control, which have promising applications in many fields including microscopy, particle manipulation, quantum science, and optical communication.

## Methods

### Experimental details

The designed metasurfaces consist of gold nanorods sitting on a glass substrate. A standard E-beam lithography (EBL) procedure is used to fabricate the designed metasurfaces. Initially, 120-nm-thick positive tone E-beam resist (PMMA) is spin-coated on a glass substrate and then baked on a hotplate at 185 °C for 4 min. The PMMA resist is exposed by an electron beam with an accelerating voltage of 30 kV and a beam current of 12 pA. The PMMA patterns are obtained by immersing the sample in the mixture of MIBK: IPA (1:3) for 45 s and in IPA for 45 s for rinsing. Afterwards, an E-beam evaporator is used to deposit a 40-nm-thick gold film on the sample. Finally, the gold nanorods are ready for characterization after the lift-off process in acetone for 6 h.

## Data availability

The data that support the findings of this study are available from the corresponding author upon request.

## Code availability

The code used in the present work is available from the corresponding author upon request.

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

## Acknowledgements

X.C. acknowledges funding support from Engineering and Physical Sciences Research Council (EP/P029892/1), Leverhulme Trust (RPG-2021-145), Royal Society International Exchanges (IES\R3\193046), and National Natural Science Foundation of China (NSFC)-Royal Society of Edinburgh (RSE) joint project.

## Author contributions

X.C. and H.A. initiated the idea. H.A. conducted the numerical simulations. H.A. fabricated the samples. H.A., M.A.A., and Y.L. performed the measurements. H.A., X.C., M.A.A., and T.Z. prepared the manuscript. X.C. supervised the project with some input from T.Z. and M.Q.M. All the authors discussed and analyzed the results.

## Competing interests

The authors declare no competing interests.
