## [Peer Review File · Nature Communications]

REVIEWER COMMENTS

Reviewer #1 (Remarks to the Author):

The manuscript entitled « Dynamic Control of Hybrid Grafted Perfect Vector Vortex Beams » by Hammad Ahmed, Muhammad Afnan Ansari, Yan Li, Thomas Zentgraf, Muhammad Qasim Mehmood and Xianzhong Chen, manuscript NCOMMS-22-51438, submitted for publication in nature communications has been reviewed.

In this manuscript the authors report on the generation of optical beams with spatially variant phase and polarization profiles. They report on a new sort of vectorial vortex beams achieved using a superposition of multiple hybrid grafter perfect vortex beam. The beams are generated using a metasurface (MS) cascaded behind a wave plate. The transmitted fields can be dynamically addressed by rotating the polarization of the incident light with a half waveplate. Rotating the waveplate modifies the input field polarization on the MS component, which results in tunable behavior of the compound system. The results are interesting but the applicative interest these "new" beams is to my appreciation relatively marginal. I noticed several points that are worth investigating further to improve the readability of this manuscript.

A detailed discussion on the diffraction efficiency is needed to better highlight the limitation of the plasmonic structures for this proof of concept.

A discussion explaining the design of the plasmonic rods is needed.

In the concluding remarks, the authors said that these beams have promising applications in microscopy, quantum science, optical communication and particle manipulation. The motivation of considering complex beams for this list of applications is not obvious.

Reviewer #2 (Remarks to the Author):

In this manuscript, Ahmed et al. demonstrate an original methodology to generate and control vector vortex beams with uniquely defined features that are design defined. The authors introduce a new type of beam, called the hybrid grafted perfect vector vortex beam (GPVVB), which is generated using a multifunctional metasurface and features a varying polarization depending on the transmission axis. Additionally, the team demonstrates the ability to dynamically control the polarization of these beams using a half wave plate (HWP), allowing for manipulation of the polarization profile through rotation of the HWP, thereby enable a number of applications ranging from particle manipulation in an optical tweezer to dense data communication, showcasing significance to the field.

With the below mentioned minor revisions and clarification, this work may be publishable Nature Communications.

1. In the results section on page 3, “When a linearly polarized light beam impinges onto the metasurface ...”. Here, the authors are encouraged to highlight to the readers that the impinging linearly polarized light beam is a superposition of the two circular polarization states.
2. The metasurface demonstrated by the authors is described to consist of gold nanorods. What is the achieved transmission using this structure? Since the design wavelength is 633nm, why is TiO₂ not considered for the material of the nanorods?
3. The geometry of the gold nanorods is highlighted on page 5. Is this optimized for this work or based on a previous work? If it's the former, please briefly indicate how the shape birefringence of the structures was optimized (i.e. studying the polarization conversion efficiency, or grating efficiency, etc) - this can be added as a short section in the supplementary material. If it is the latter, please reference the work used within the manuscript.

4. A suggestion to the authors: use abbreviations where possible in the equations to add simplicity. For example, in Equation (3), replacing φ_{axicon} with φ_{ax} .

5. The phase profile to generate an axicon is $\varphi(x, y) = -\frac{2\pi}{\lambda} \cdot \sqrt{x^2 + y^2} \cdot NA$,

where NA is the numerical aperture. The NA is also: $NA = \frac{\lambda}{2u}$, where u is the

axicon period. Therefore the phase profile for the axicon can be rewritten as:

$\varphi(x, y) = -\frac{\pi}{u} \cdot \sqrt{x^2 + y^2}$. Please double check this equation and confirm the

phase profile on page 4 and update the equation as required.

6. How are the designed devices simulated? Is the entire device simulated using FDTD? The authors are encouraged to discuss and elaborate on the simulation approach used.

7. Why is there a discrepancy in the simulation compared to the theory, as show in S7? The authors are encouraged to further discuss this and outline the causes that lead to this difference.

8. Please add the following to the figures:

i. In Figure (2c), in order to make a visually easy comparison between the simulation and experiment, please use the same colorbar.

ii. In Figures (3) and (4), in addition to using the same colorbar, also add the colorbar within the figure, even if it is the same as that in Figure 2.

iii. In Figure (4a), for the experiment column, what do the white arrows really mean here? Are these measured polarizations? Meaning, are these back calculated from the measurement intensity profile or just an overlay of theory/simulation? In either case, please outline this in the figure description.

Reviewer #3 (Remarks to the Author):

Ahmed H. et al. proposed a hybrid grafted perfect vector vortex beams (GPVVBs) using gold nanorod composed metasurface by superposing multiple hybrid grafted perfect vortex beams (GPVBs). Compared to previous research of the authors (*Adv. Mater.* 2022, 34, 2203044), here the authors successfully demonstrated a spatially variant polarization profile of GPVVBs. The authors proposed a metasurface concept that generates different polarization orders in the 2D domain in the vortex beam and the effective manipulation of the polarization of generated GPVVBs by simply rotating the HWP of the output beam. Furthermore, the authors explored different rotation speeds of the lobes in each polarization sector of high-order GPVVBs. These additional degrees of freedom of the GPVVBs could advance the field of dynamic optical tweezers, optical communications, and optical data storage with extra information capacity. The manuscript is well-written and the simulation and experimental data are well-shown. Therefore, I would like to recommend publication after the points noted below have been addressed with appropriate revision. See the detailed comments below.

1. In Fig. 2c, at the 1st and 2nd row, the author demonstrated GPVVBs with polarization order $m_1 = -1$, and $m_2 = 2$ without analyzer. I understand that the simulated intensity profile would have an equal distribution, however, the intensity profile of the experimental data seems unequal. The authors should provide an intensity profile depending on the orientation angle and discuss the possible factors that could affect the intensity profiles of GPVVBs.

2. In the manuscript, the authors demonstrated hybrid GPVVBs by composing two higher-order GPVVBs with different topological charges using circularly polarized light RCP and LCP. However, I am curious about the limits of superposing different GPVVBs on a single metasurface.

3. I am afraid of the word dynamic tuning, and the phrase 'making the metasurface function as a dynamic optical device' since the control of the polarization is by the HWP placed between the metasurface and the analyzer, controlling the output beam of the metasurface.

4. A small ring profile exists inside the inner ring with polarization $m_1 = -4$, and $m_2 = -2$ of hybrid GPVVBs in Fig. 4a. Could the authors discuss this feature?

5. The authors should provide the conversion efficiency of the proposed metasurface, by calculating the unit gold nanorod. In addition, the authors should discuss the efficiency of the metasurface.

6. Several important references to optical metasurfaces describing multifunctional metasurfaces that generate or manipulate vortex beams are missing.

Chem. Rev. 2021, 121, 21, 13013–13050

Adv. Mater. 2022, 2106692

Nanoscale, 2022, 14, 4380-4410

7. Some minor comments

- Graphs in Fig. 3c and 3d are hardly readable. Some modifications for readability are required.
- I believe GPVBs is appropriate except for GVBs at line 61.
- Italic at μm , nm.
- Journal abbreviation in reference.

Response to reviewers' comments

We thank all the reviewers for their positive comments. Their valuable suggestions are greatly helpful for us to improve our manuscript. In the following, we have provided a point-to-point reply to the reviewers' concerns and raised questions. The changes are highlighted in red color in the manuscript. We hope that the revised paper can meet the journal's criteria for publication.

COMMENTS TO AUTHOR:

Reviewer #1:

The manuscript entitled « Dynamic Control of Hybrid Grafted Perfect Vector Vortex Beams » by Hammad Ahmed, Muhammad Afnan Ansari, Yan Li, Thomas Zentgraf, Muhammad Qasim Mehmood and Xianzhong Chen, manuscript NCOMMS-22-51438, submitted for publication in nature communications has been reviewed.

In this manuscript the authors report on the generation of optical beams with spatially variant phase and polarization profiles. They report on a new sort of vectorial vortex beams achieved using a superposition of multiple hybrid grafter perfect vortex beam. The beams are generated using a metasurface (MS) cascaded behind a wave plate. The transmitted fields can be dynamically addressed by rotating the polarization of the incident light with a half waveplate. Rotating the waveplate modifies the input field polarization on the MS component, which results in tunable behavior of the compound system.

The results are interesting but the applicative interest these “new” beams is to my appreciation relatively marginal. I noticed several points that are worth investigating further to improve the readability of this manuscript.

Reply: We thank the reviewer for the positive comment and valuable suggestions.

1. A detailed discussion on the diffraction efficiency is needed to better highlight the limitation of the plasmonic structures for this proof of concept.

Reply: We thank the reviewer for the valuable question. We have provided simulated and measured diffraction efficiency. The diffraction efficiency (η_D) can be defined as the ratio of the power of the diffracted light (I_D) to the total power of the transmitted light through the metasurface (I_T). I_D is the same as the power of cross-polarised light due to the off-axis design. Moreover, I_T is the sum of the power of the co-polarised light (I_{co}) and that of the cross-polarised light (I_{cross}). Mathematically, η_D can be expressed as^{7, 8, 9}:

$$\eta_D = \frac{I_D}{I_T} \approx \frac{I_{cross}}{I_{co} + I_{cross}}. \quad (S14)$$

The simulated and measured diffraction efficiencies in our work are about 34.3% and 13.9% at 633 nm, respectively. We also measured the diffraction efficiency at different wavelengths as illustrated in **Fig. S3**. As a proof-of-concept, we used plasmonic metasurfaces with low efficiency, which can be dramatically improved with dielectric metasurfaces.

To clarify this point in the main manuscript, we added “The details about the efficiency of metasurfaces are provided in **Supplementary Section 4**.” (Line 127-128, Page 5)

In Supplementary Section 4, we added:

“The diffraction efficiency (η_D) can be defined as the ratio of the power of diffracted light (I_D) to the total power of transmitted light through the metasurface (I_T). Since the off-axis design is used here, I_D is the same as the power of cross-polarised light. Therefore, I_T is the sum of the power of the co-polarised light (I_{co}) and that of the cross-polarised light (I_{cross}). Mathematically, η_D can be expressed as ^{7,8,9}:

$$\eta_D = \frac{I_D}{I_T} \approx \frac{I_{cross}}{I_{co} + I_{cross}} \quad (S14)$$

The simulated and measured diffraction efficiencies for the proposed work are 34.3% and 13.9% at 633 nm, respectively. We also measure the diffraction efficiency at other wavelengths as illustrated in **Fig. S3**. As a proof-of-concept, we use plasmonic metasurfaces with a low efficiency, which can be dramatically improved using dielectric metasurfaces.”

In Supplementary Section 4, we added the following figure.

Fig. S3. Simulated and measured diffraction efficiency

The following references are added in the Supplementary Information:

7. Zhou Z, Li J, Su R, Yao B, Fang H, Li K, et al. Efficient silicon metasurfaces for visible light. *Acs Photonics* 2017, 4(3): 544-551.

8. Lin D, Fan P, Hasman E, Brongersma ML. Dielectric gradient metasurface optical elements. *Science* 2014, 345(6194): 298-302.
9. Deng Z-L, Deng J, Zhuang X, Wang S, Li K, Wang Y, et al. Diatomic metasurface for vectorial holography. *Nano Letters* 2018, 18(5): 2885-2892.

2. *A discussion explaining the design of the plasmonic rods is needed.*

Reply: We thank the reviewer for the valuable suggestion. To clarify this point in the manuscript, we added “The inset of **Fig. 2(b)** shows an SEM image of the fabricated metasurface sample, which consists of gold (Au) nanorods with various rotation angles ϕ on a glass (SiO₂) substrate. The geometry of nanorods is based on our previous work³³. Each nanorod is 0.08 μm wide, 0.2 μm long, and 0.04 μm thick. The sample has an area of $200 \times 200 \mu\text{m}^2$ with a pixel size of $0.3 \times 0.3 \mu\text{m}^2$. Detailed information about the unit cell design is provided in **Supplementary Section 3**. The nanofabrication details are given in the Methods section.” (Line 121-127, Page 5)

To clarify this point, in Supplementary Section 3, we added:

“The proposed devices are realized based on plasmonic metasurfaces, which consist of gold (Au) nanorods with spatially varying orientations on a glass substrate as shown in **Fig. S2**. Such nanorods show strong light-matter interaction when they are resonantly excited at their localized surface plasmon polariton resonance. The length, width, and height of each nanorod are $L = 0.2 \mu\text{m}$ long, $W = 0.08 \mu\text{m}$, and $H = 0.2 \mu\text{m}$, respectively. The periodicity (P) of each unit cell is $P = 0.3 \mu\text{m}$ along both the x and y directions. The experimental and simulated efficiencies are illustrated in **Supplementary Section 4**. The simulated efficiency is calculated using the frequency domain solver of the Computer Simulation Technology (CST) Microwave Studio software. In the simulation, the permittivity of the gold nanorods is obtained by a Drude model with the plasma frequency $\omega_p = 1.37 \times 10^{16}$ rad/s, and the collision frequency $\gamma_c = 1.215 \times 10^{14}$ rad/s. The refractive index of the substrate is 1.46. The unit cell boundary is used along both the x and y directions, while the open boundary condition is used along the z direction. Although the cross-polarised part is low and the co-polarised part is high, the two parts are separated using the off-axis design as shown in the experimental setup in **Fig. S6**. Dielectric metasurfaces can be used to significantly improve the efficiency.

Fig. S2. Unit cell design. (a) Schematic of Au nanorods on a glass substrate. Length ($L = 0.2 \mu\text{m}$), width ($W = 0.08 \mu\text{m}$), height ($H = 0.2 \mu\text{m}$), and periodicity ($P = 0.3 \mu\text{m}$) are geometric parameters. (b) Top view of the unit cell with a nanorod rotated at an angle ϕ .

3. *In the concluding remarks, the authors said that these beams have promising applications in microscopy, quantum science, optical communication and particle manipulation. The motivation of considering complex beams for this list of applications is not obvious.*

Reply: We appreciate the reviewer for the critical point. The uniqueness of the method and exotic properties of the novel beams may find applications in many related research fields. For example, optical trapping can hold and rotate delicate biological samples, leading to the improvement in the 3D microscopic imaging of live cells by imaging samples from different directions ^{41, 42}. However, existing techniques for holding and orienting cells have used complex optical apparatus ⁴³ and have only been applied to single cells or small clusters of cells, limiting the practical applications within the community of cell biologists. Moreover, the proposed metasurface approach can trap multiple particles and provide more degrees of freedom to rotate the trapped biological structures. The availability of more TCs in a single beam and the dynamic control of VVBs can allow researchers in quantum science and optical communications to encode more information in the light beams, which will dramatically increase the information capacity and offer more design flexibility.

To clarify this point in the manuscript, we added “The uniqueness of the method and exotic properties of the novel beams may find applications in many related research fields. For example, optical trapping can hold and rotate delicate biological samples, leading to the improvement in the 3D microscopic imaging of live cells by imaging samples from different directions ^{41, 42}. However, existing techniques for holding and orienting cells have used complex optical apparatus ⁴³ and have only been applied to single cells or small clusters of cells, limiting the practical applications within the community of cell biologists. Moreover, the proposed metasurface approach can trap multiple particles and provide more degrees of

freedom to rotate the trapped biological structures. The availability of more TCs in a single beam and the dynamic control of VVBs can allow researchers in quantum science and optical communications to encode more information in the light beams, which will dramatically increase the information capacity and offer more design flexibility.” (Line 286-296, Page 10)

The following references are added in the manuscript:

41. Kolb T, Albert S, Haug M, Whyte G. Dynamically reconfigurable fibre optical spanner. *Lab on a Chip* 14, 1186-1190 (2014).

42. Kreysing M, et al. Dynamic operation of optical fibres beyond the single-mode regime facilitates the orientation of biological cells. *Nature communications* 5, 5481 (2014).

43. Kolb T, Albert S, Haug M, Whyte G. Optofluidic rotation of living cells for single-cell tomography. *Journal of biophotonics* 8, 239-246 (2015).

Reviewer #2:

In this manuscript, Ahmed et al. demonstrate an original methodology to generate and control vector vortex beams with uniquely defined features that are design defined. The authors introduce a new type of beam, called the hybrid grafted perfect vector vortex beam (GPVVB), which is generated using a multifunctional metasurface and features a varying polarization depending on the transmission axis. Additionally, the team demonstrates the ability to dynamically control the polarization of these beams using a half wave plate (HWP), allowing for manipulation of the polarization profile through rotation of the HWP, thereby enable a number of applications ranging from particle manipulation in an optical tweezer to dense data communication, showcasing significance to the field. With the below mentioned minor revisions and clarification, this work may be publishable Nature Communications.

Reply: We thank the reviewer for the positive comments.

1. *In the results section on page 3, “When a linearly polarized light beam impinges onto the metasurface ...”. Here, the authors are encouraged to highlight to the readers that the impinging linearly polarized light beam is a superposition of the two circular polarization states.*

Reply: We appreciate the reviewer for the valuable suggestion. To clarify this point in the manuscript, we added “A linear polarization state can be decomposed into two different circular polarization states.” (Line 79-80, Page 3)

2. *The metasurface demonstrated by the authors is described to consist of gold nanorods. What is the achieved transmission using this structure? Since the design wavelength is 633nm, why is TiO2 not considered for the material of the nanorods?*

Reply: We thank the reviewer for the very good point. We have provided simulated and experimentally measured transmission efficiencies of the proposed design. The transmission efficiency is the ratio of total transmitted power through the metasurface (I_T) to the input power (I_{in}). I_T is the sum of power of co-polarised light (I_{co}) and the power of cross-polarised light (I_{cross}). The mathematical expression can be written as ^{7,8,9}:

$$\eta_T = \frac{I_T}{I_{in}} \approx \frac{I_{co} + I_{cross}}{I_{in}} \quad (S15)$$

The simulated and measured transmission efficiencies (**Fig. S4**) at 633 nm are 52.7% and 51%, respectively. The overall curves are relatively flat and uniform within the broadband ranging from 500 nm to 700 nm. It is worth noting here that the simulated and measured converted transmission efficiencies are 18% and 7%, respectively. For better understanding, we also plot I_{co} and I_{cross} separately in **Fig. S5**.

As a proof of concept, we used plasmonic metasurfaces consisting of gold nanorods with a low conversion efficiency. We agree with the reviewer that dielectric metasurfaces with TiO2 nanopillars can dramatically improve efficiency in visible domain.

To clarify this point in manuscript, we added “The details about the efficiency of metasurface are provided in **Supplementary Section 4**.” (Line 127-128, Page 5)

To clarify this point, in **Supplementary Section 4**, we added: “The transmission efficiency is the ratio of total transmitted power through the metasurface (I_T) to the input power (I_{in}). I_T is the sum of the power of co-polarised light (I_{co}) and that of cross-polarised light (I_{cross}). The mathematical expression can be written as ^{7,8,9}:

$$\eta_T = \frac{I_T}{I_{in}} \approx \frac{I_{co} + I_{cross}}{I_{in}} \quad (S15)$$

The simulated and measured transmission efficiencies (**Fig. S4**) at 633 nm are 52.7% and 51%, respectively. The overall curves are relatively flat and uniform within the broadband

ranging from 500 nm to 700 nm. It is worth noting here that the simulated and measured converted transmission efficiencies are 18% and 7%, respectively. For better understanding, we also plot I_{co} and I_{cross} separately in Fig. S5.”

Fig. S4. Simulated and measured transmission efficiency.

Fig. S5. Simulated and measured co-polarisation and cross-polarisation efficiency.

The following references are added to the supplementary information:

7. Zhou Z, Li J, Su R, Yao B, Fang H, Li K, et al. Efficient silicon metasurfaces for visible light. *ACS Photonics* 2017, 4(3): 544-551.
 8. Lin D, Fan P, Hasman E, Brongersma ML. Dielectric gradient metasurface optical elements. *Science* 2014, 345(6194): 298-302.
 9. Deng Z-L, Deng J, Zhuang X, Wang S, Li K, Wang Y, et al. Diatomic metasurface for vectorial holography. *Nano letters* 2018, 18(5): 2885-2892.
3. *The geometry of the gold nanorods is highlighted on page 5. Is this optimized for this work or based on a previous work? If it's the former, please briefly indicate how the shape birefringence of the structures was optimized (i.e. studying the polarization conversion*

efficiency, or grating efficiency, etc) - this can be added as a short section in the supplementary material. If it is the latter, please reference the work used within the manuscript.

Reply: The geometry of the gold nanorods is based on our previous work. We thank the reviewer for the good suggestion and have cited the reference as follows: “The geometry of nanorod is based on our previous work³³. Each nanorod is 0.08 μm wide, 0.2 μm long, and 0.04 μm thick.” (Line 123-124, Page 5)

The following reference is added to the manuscript:

33. Wang R, et al. Metalens for generating a customized vectorial focal curve. *Nano Letters* **21**, 2081-2087 (2021).

4. A suggestion to the authors: use abbreviations where possible in the equations to add simplicity. For example, in Equation (3), replacing with φ_{axicon} with φ_{ax} .

Reply: We appreciate the reviewer for the valuable suggestion. We have updated the equations by replacing φ_{axicon} with φ_{ax} .

5. The phase profile to generate an axicon is $\varphi(x, y) = -\frac{2\pi}{\lambda}\sqrt{x^2 + y^2} \cdot NA$, where NA is the numerical aperture. The NA is also: $\frac{\lambda}{2u}$, where u is the axicon period. Therefore, the phase profile for the axicon can be rewritten as: $\varphi(x, y) = -\frac{\pi}{u}\sqrt{x^2 + y^2}$. Please double check this equation and confirm the phase profile on page 4 and update the equation as required.

Reply: We apologize for the typo and thank the reviewer for the valuable point. The equation has been updated in the manuscript.

6. How are the designed devices simulated? Is the entire device simulated using FDTD? The authors are encouraged to discuss and elaborate on the simulation approach used.

Reply: We thank the reviewer for the valuable question. All the designed devices are simulated using the Kirchhoff diffraction integration. The actual size of the metasurfaces is used to calculate the far-field intensity profiles by the following equation^{10, 11}:

$$E(x, y) = \frac{e^{ikz}}{i\lambda z} e^{\frac{ik}{2z}(x^2+y^2)} \iint E(x_0, y_0) e^{-\frac{ik}{z}(xx_0+yy_0)} dx_0 dy_0 \quad (S16)$$

where $E(x_0, y_0)$ is the complex amplitude profile of GPVVB at the $z = 0$. x_0 and y_0 are the coordinates of nanorods, while x and y are the coordinates of the observation plane

at a distance of z . $k = \frac{2\pi}{\lambda}$ is the wavevector and λ denotes wavelength. In the simulation, the amount of time needed for the calculation depends on the resolution of the observation plane (x, y) . In this work, the area of $200 \times 200 \mu\text{m}^2$ on the observation plane with a resolution of $0.3 \mu\text{m}$ is used in our simulation.

To clarify this point, in the manuscript we added “The simulation of designed metasurfaces is performed based on the Kirchhoff diffraction integration¹¹, the details can be found in **Supplementary Information Section 6.**” (Page 5, Line 134-135)

In Supplementary Information, we added more details as follows:

“Supplementary Section 6. Kirchhoff diffraction integration

All the designed devices are simulated using the Kirchhoff diffraction integration. The actual size of the metasurfaces is used to calculate the far-field intensity profiles by the following equation^{10, 11}:

$$E(x, y) = \frac{e^{ikz}}{i\lambda z} e^{\frac{ik}{2z}(x^2+y^2)} \iint E(x_0, y_0) e^{-\frac{ik}{z}(xx_0+yy_0)} dx_0 dy_0$$

where $E(x_0, y_0)$ is the complex amplitude profile of GPVVB at the $z = 0$. x_0 and y_0 are the coordinates of nanorods, while x and y are the coordinates of the observation plane at a distance of z . $k = \frac{2\pi}{\lambda}$ is the wavevector and λ denotes wavelength. In the simulation, the amount of time needed for the calculation depends on the resolution of the observation plane (x, y) . In this work, the area of $200 \times 200 \mu\text{m}^2$ on the observation plane with the resolution of $0.3 \mu\text{m}$ is used in our simulation.

The following references are added in the Supplementary Information:

10. Intaravanne Y, Wang R, Ahmed H, Ming Y, Zheng Y, Zhou Z-K, et al. Color-selective three-dimensional polarization structures. *Light: Science & Applications* 2022, 11(1): 302.

11. Zhang Y, Liu W, Gao J, Yang X. Generating focused 3D perfect vortex beams by plasmonic metasurfaces. *Advanced Optical Materials* 2018, 6(4): 1701228.

7. *Why is there a discrepancy in the simulation compared to the theory, as show in S7? The authors are encouraged to further discuss this and outline the causes that lead to this difference.*

Reply: We thank the reviewer for raising this point. Theoretical rotation angle values are calculated based on **Eq. 4**

$$\theta_n = \frac{2\alpha}{Nm_n}, \quad (4)$$

While the simulated rotation angles are acquired using Kirchhoff diffraction integration. The slight discrepancy between theory and simulation is mainly due to the resolution of the observation plane in the Kirchhoff diffraction integration. The simulated rotation angles can approach theoretical values by increasing the resolution. However, the time for simulation and computation will increase.

To clarify this point in the manuscript we added “The slight discrepancy between theory and simulation is mainly due to the resolution of the observation plane in the Kirchhoff diffraction integration. The simulated rotation angles can approach theoretical values by increasing the resolution.” (Page 7, Line 184-186)

8. Please add the following to the figures:

I. In Figure (2c), in order to make a visually easy comparison between the simulation and experiment, please use the same colorbar.

Reply: We thank the reviewer for the valuable suggestion and have updated the colorbars.

II. In Figures (3) and (4), in addition to using the same colorbar, also add the colorbar within the figure, even if it is the same as that in Figure 2.

Reply: We thank the reviewer for the valuable suggestion and have updated the colorbars for all the figures in the manuscript as well as in supplementary information.

III. In Figure (4a), for the experiment column, what do the white arrows really mean here? Are these measured polarizations? Meaning, are these back-calculated from the measurement intensity profile or just an overlay of theory/simulation? In either case, please outline this in the figure description.

Reply: We thank the reviewer for the valuable question. In Figure (4a), the white arrows represent the measured polarization states. In the experiment, the polarization state of the output light beam is obtained using Stokes polarimetry^{1, 15}. By measuring the series of intensity profiles by adding an analyser, the stokes parameters can be calculated before the camera. The Stokes parameters can be expressed as^{xx}:

$$S_0 = I(0^\circ) + I(90^\circ) \quad S17$$

$$S_1 = I(0^\circ) - I(90^\circ) \quad S18$$

$$S_2 = I(45^\circ) - I(135^\circ) \quad S19$$

Here $I(0^\circ)$, $I(45^\circ)$, $I(90^\circ)$, $I(135^\circ)$ are the intensities of GPVVBs after an analyser rotated at 0° , 45° , 90° , 135° with respect to the x – axis. The spatial distribution of polarization states as shown in Fig. 4a can be calculated as¹³:

$$J = \frac{1}{2} \arctan\left(\frac{s_2}{s_1}\right) \quad S20$$

To clarify this point, in the manuscript we added “Blue double arrows and white arrows represent the linear polarization direction of incident light and that on the hybrid GPVVB, respectively. The spatial polarization profiles are calculated based on the Stokes polarimetry^{1, 15}. Details are provided in **Supplementary Information Section 11.**” (Page 7, Line 204-207)

In Supplementary Information, we added more details as follows:

“Supplementary Section 11. Polarization Measurement

The polarization state of the output light beam is obtained using the Stokes polarimetry^{1, 12}. By measuring the series of intensity profiles by adding an analyser, the stokes parameters can be calculated. The Stokes parameters can be expressed as¹³:

$$S_0 = I(0^\circ) + I(90^\circ) \quad S17$$

$$S_1 = I(0^\circ) - I(90^\circ) \quad S18$$

$$S_2 = I(45^\circ) - I(135^\circ) \quad S19$$

Here $I(0^\circ)$, $I(45^\circ)$, $I(90^\circ)$, $I(135^\circ)$ are the intensities of GPVVBs after an analyser rotated at 0° , 45° , 90° , 135° with respect to the x axis. The spatial distribution of polarization states as shown in Fig. 4a can be calculated as¹³:

$$J = \frac{1}{2} \arctan\left(\frac{s_2}{s_1}\right) \quad S20$$

The following references are added in the Supplementary Information:

“12. Song Q, Baroni A, Wu PC, Chenot S, Brandli V, Vézian S, et al. Broadband decoupling of intensity and polarization with vectorial Fourier metasurfaces. Nature communications 2021, 12(1): 3631.

13. Liu M, Huo P, Zhu W, Zhang C, Zhang S, Song M, et al. Broadband generation of perfect Poincaré beams via dielectric spin-multiplexed metasurface. Nature communications 2021, 12(1): 2230.”

Reviewer #3:

Ahmed H. et al. proposed a hybrid grafted perfect vector vortex beams (GPVVBs) using gold nanorod composed metasurface by superposing multiple hybrid grafted perfect vortex beams

(GPVVBs). Compared to previous research of the authors (*Adv. Mater.* 2022, 34, 2203044), here the authors successfully demonstrated a spatially variant polarization profile of GPVVBs. The authors proposed a metasurface concept that generates different polarization orders in the 2D domain in the vortex beam and the effective manipulation of the polarization of generated GPVVBs by simply rotating the HWP of the output beam. Furthermore, the authors explored different rotation speeds of the lobes in each polarization sector of high-order GPVVBs. These additional degrees of freedom of the GPVVBs could advance the field of dynamic optical tweezers, optical communications, and optical data storage with extra information capacity. The manuscript is well-written and the simulation and experimental data are well-shown. Therefore, I would like to recommend publication after the points noted below have been addressed with appropriate revision. See the detailed comments below.

Reply: We thank the reviewer for the very positive comments.

1. In Fig. 2c, at the 1st and 2nd row, the author demonstrated GPVVBs with polarization order $m_1 = -1$, and $m_2 = 2$ without analyzer. I understand that the simulated intensity profile would have an equal distribution, however, the intensity profile of the experimental data seems unequal. The authors should provide an intensity profile depending on the orientation angle and discuss the possible factors that could affect the intensity profiles of GPVVBs.

Reply: We thank the reviewer for the good point and have provided the normalized intensity profile. We take the intensity profile of one polarization profile as a reference and then normalize the other intensities with respect to the reference. The reason for unequal intensities in the experimental data is mainly due to the imperfection of the optical setup (e.g., inaccurate alignment) and sample quality (e.g., missing nanorods).

To clarify this point in the manuscript we added “The unequal intensities in the experimental data without polarizer are mainly due to the imperfection of the optical setup (e.g., inaccurate alignment) and sample quality (e.g., missing nanorods).” (Page 5, Line 140-142)

2. In the manuscript, the authors demonstrated hybrid GPVVBs by composing two higher-order GPVVBs with different topological charges using circularly polarized light RCP and LCP. However, I am curious about the limits of superposing different GPVVBs on a single metasurface.

Reply: We thank the reviewer for the very interesting question. In our experiment, we have demonstrated the hybrid GPVVB that is composed of two higher-order GPVVBs. The experimental results agree well with the simulation results. Although the design procedure for the hybrid GPVVB with more than two higher-order GPVVBs is almost the same, there are some limitations in both the numerical optimization process and fabrication. The maximum number of higher-order GPVVBs involved is mainly determined by the sample size and the design parameters of metasurfaces (e.g., pixel size). Generally speaking, the increase in pixel number and the reduction of the pixel size of the metasurface can increase the number of GPVVBs. As an example, we simulated and experimentally realized a $400\mu\text{m} \times 400\mu\text{m}$ metasurface with four higher-order GPVVBs. The fabrication details and results (simulation and experimental) are provided in **Fig S7 and S11**, respectively.

To clarify this point, in the manuscript, we added “The maximum number of higher-order GPVVBs involved is mainly determined by the sample size and the design parameters of metasurfaces (e.g., pixel size). For instance, the increase in pixel number and the reduction of the pixel size of the metasurface can increase the number of GPVVBs. As an example, we simulate and experimentally realize a $400 \times 400 \mu\text{m}^2$ metasurface with four higher-order GPVVBs. The fabrication details and results (simulation and experimental) are provided in **Fig. S7 and Fig. S11**, respectively.” (Page 10, Line 274-280) In Supplementary Information, we added more details as follows:

“Supplementary Section 7. Metasurface Design

Fig. S7. (f) Left: Phase profile for Hybrid GPVVBs with polarization orders: First ring: $m_1 = -4, m_2 = -2$, second ring: $m_1 = +3, m_2 = +6, m_3 = +9$ third ring: $m_1 = -4, m_2 = -2$ and fourth ring: $m_1 = +3, m_2 = +6, m_3 = +9$. **Right:** SEM image of the fabricated sample.

“Supplementary Section 10. Hybrid GPVVB Generation and Manipulation

Fig. S11. Hybrid GPVVB Generation with four higher order GPVVBs. (a) Intensity profile and polarization distribution of a hybrid GPVVB under the illumination of linearly polarized light along the horizontal direction. Blue double arrows and white arrows show the linear polarization direction of incident light and that of the hybrid GPVVB, respectively. (b) Modulated intensity profiles after passing through an analyser. The rotation of lobes in different directions indicates the existence of various polarization orders. The rotation directions of the rings are shown in green and white curved arrows, respectively. The first and third ring is formed by grafting two OVs while the second and fourth ring is formed by grafting three OVs. Here the axicon periods for four rings are $u_1 = 4 \mu\text{m}$, $u_2 = 3.3 \mu\text{m}$, $u_3 = 2.8 \mu\text{m}$ and $u_4 = 2.5 \mu\text{m}$, respectively”.

3. *I am afraid of the word dynamic tuning, and the phrase ‘making the metasurface function as a dynamic optical device’ since the control of the polarization is by the HWP placed between the metasurface and the analyzer, controlling the output beam of the metasurface.*

Reply: We apologize for the confusion and have replaced the word “dynamic tuning” with “dynamic control” to avoid confusion. Moreover, we have also replaced the phrase “making the metasurface function as a dynamic optical device” with “making overall optical system dynamically controllable.”

To clarify this point, in the manuscript, we added “Furthermore, these beams are dynamically controlled with a rotating half waveplate, making an overall optical system dynamically controllable.” (Page 1, Line 28-30)

4. *A small ring profile exists inside the inner ring with polarization $m_1 = -4$, and $m_2 = -2$ of hybrid GPVVBs in Fig. 4a. Could the authors discuss this feature?*

Reply: We thank the reviewer for the raising this point. The existence of small ring with very weak signal is due to the parasitic light. The parasitic light arises from the crosstalk between the rings. Similar problem has been encountered by other groups while generating conventional perfect vortices. For example,

<https://opg.optica.org/ol/fulltext.cfm?uri=ol-39-18-5305&id=300733>

<https://opg.optica.org/ol/fulltext.cfm?uri=ol-38-4-534&id=249164>

<https://iopscience.iop.org/article/10.1088/1361-6463/ac8d13/meta>

However, in this case, we notice that it can be minimized by meticulously optimizing the axicon period and the size of the metasurface. As an example, we simulate and experimentally realize a $400\mu\text{m} \times 400\mu\text{m}$ metasurface with four higher order GPVVBs. Here we carefully choose the four different axicon periods ($u_1 = 4\ \mu\text{m}$, $u_2 = 3.3\ \mu\text{m}$, $u_3 = 2.8\ \mu\text{m}$ and $u_4 = 2.5\ \mu\text{m}$) to mitigate the small ring effect. The simulation and experimental results are provided in **Fig S11**. It can be seen that by increasing the size of metasurface and carefully choosing the axicon period, the existence of small ring can be minimized.

Fig. S11. Hybrid GPVVB Generation with four higher order GPVVBs.

To clarify this point, in the manuscript, we added “The existence of small ring with very weak intensity is due to the parasitic light, which comes from the crosstalk between two rings.^{34, 35, 36} The parasitic light can be minimized by optimizing the axicon period and the size of the metasurface. As an example, we simulate and experimentally realize a metasurface with an area of $400 \times 400\ \mu\text{m}^2$ to generate four higher-order GPVVBs. The small ring effect is dramatically suppressed. The simulation and experimental results are provided in **Fig. S11**” (Page 8, Line 208-214)

The following references are added in the manuscript:

34. García-García J, Rickenstorff-Parrao C, Ramos-García R, Arrizón V, Ostrovsky AS. Simple technique for generating the perfect optical vortex. *Optics letters* 39, 5305-5308 (2014).
35. Ostrovsky AS, Rickenstorff-Parrao C, Arrizón V. Generation of the “perfect” optical vortex using a liquid-crystal spatial light modulator. *Optics letters* 38, 534-536 (2013).
36. Long Z, Hu H, Ma X, Tai Y, Li X. Encoding and decoding communications based on perfect vector optical vortex arrays. *Journal of Physics D: Applied Physics* 55, 435105 (2022).
5. *The authors should provide the conversion efficiency of the proposed metasurface, the unit gold nanorod. In addition, the authors should discuss the efficiency of the metasurface.*

Reply: We thank the reviewer for the good point. We have provided simulated and measured conversion efficiencies in our design. The conversion efficiency is the ratio of cross-polarised power through the metasurface (I_{cross}) to the input power (I_{in}). The mathematical expression can be written as ^{7,8,9}:

$$\eta_T = \frac{I_{cross}}{I_{in}}$$

The simulated and measured conversion efficiency at 633 nm (**Fig. S5**) is about 18% and 7%, respectively. We also measured the conversion efficiency at different wavelengths as illustrated in **Fig. S5**. Moreover, the total transmission efficiency, which is the sum of the power of co-polarised light (I_{co}) and that of cross-polarised light (I_{cross}), is provided in **Fig. S4**

To clarify this point in main manuscript we added “The details about the efficiency of metasurfaces are provided in **Supplementary Section 4**.” (Line 127-128, Page 5)

To clarify this point, in **Supplementary Section 4**, we added: “The transmission efficiency is the ratio of total transmitted power through the metasurface (I_T) to the input power (I_{in}). I_T is the sum of the power of co-polarised light (I_{co}) and that of cross-polarised light (I_{cross}). The mathematical expression can be written as ^{7,8,9}:

$$\eta_T = \frac{I_T}{I_{in}} \approx \frac{I_{co} + I_{cross}}{I_{in}} \quad (S15)$$

The simulated and measured transmission efficiencies (**Fig. S4**) at 633 nm are 52.7% and 51%, respectively. The overall curves are relatively flat and uniform within the broadband ranging from 500 nm to 700 nm. It is worth noting here that the simulated and measured converted transmission efficiencies are 18% and 7%, respectively. For better understanding, we also plot I_{co} and I_{cross} separately in **Fig. S5.**"

Fig. S4. Simulated and measured transmission efficiency.

Fig. S5. Simulated and measured co-polarisation and cross-polarisation efficiency.

The following references are added to the supplementary information:

7. Zhou Z, Li J, Su R, Yao B, Fang H, Li K, et al. Efficient silicon metasurfaces for visible light. *ACS Photonics* 2017, 4(3): 544-551.
8. Lin D, Fan P, Hasman E, Brongersma ML. Dielectric gradient metasurface optical elements. *science* 2014, 345(6194): 298-302.
9. Deng Z-L, Deng J, Zhuang X, Wang S, Li K, Wang Y, et al. Diatomic metasurface for vectorial holography. *Nano letters* 2018, 18(5): 2885-2892.
6. *Several important references to optical metasurfaces describing multifunctional metasurfaces that generate or manipulate vortex beams are missing.*

Chem. Rev. 2021, 121, 21, 13013–13050

Adv. Mater. 2022, 2106692

Nanoscale, 2022, 14, 4380-4410

Reply: We thank the reviewer for the valuable suggestion and for bringing the relevant works to our attention. In the Introduction section, we have cited these 3 references as follows: “Benefiting from the unprecedented manipulation of phase, amplitude, and polarization^{14, 15, 16, 17}, optical metasurfaces have provided a compact platform to realize various vortex beams^{18, 19, 20} such as perfect vortex beams^{11, 21}, ring vortex beams^{22, 23, 24}, OAM holography^{25, 26, 27}, vector beams^{28, 29, 30, 31}, and vector vortex beam (VVBs)^{28, 29, 30, 31}.” (Line 57-60, Page 2).

The following references are added to the supplementary information:

18. Jung C, et al. Metasurface-driven optically variable devices. *Chemical Reviews* 121, 13013-13050 (2021).

19. Ren H, Maier SA. Nanophotonic Materials for Twisted - Light Manipulation. *Advanced Materials*, 2106692 (2021).

20. Kim G, Kim S, Kim H, Lee J, Badloe T, Rho J. Metasurface-empowered spectral and spatial light modulation for disruptive holographic displays. *Nanoscale*, (2022).

7. *Some minor comments*

- *Graphs in Fig. 3c and 3d are hardly readable. Some modifications for readability are required.*

Reply: We apologize for the figure quality. We have now provided all figures in high-resolution format with 300 dpi in a separate folder along with this rebuttal letter.

- *I believe GPVBs is appropriate except for GVBs at line 61.*

Reply: We apologize for the typo. GVBs have been replaced with GPVBs.

- *Italic at μm , nm.*

Reply: μm and nm have been made non italic and updated throughout the manuscript

- *Journal abbreviation in reference.*

Reply: All the references have been updated according to the journal format.

REVIEWERS' COMMENTS

Reviewer #1 (Remarks to the Author):

The authors have considered the remarks. I believe the content is now clear and sufficient to be reproduced by others. Following on my previous comment, I still believe that an optical system operating with less than 10% of efficiency would have a hard time to find realistic application. In the future, maybe, some solution would be needed to make this approach useful.

Reviewer #2 (Remarks to the Author):

I would like to thank the authors for their detailed responses and the changes made to the manuscript and supplementary material. Please consider highlighting the possible use of TiO₂ nano pillars for this application within the manuscript - this would also encourage other researchers to explore this opportunity. Lastly, please thoroughly proofread the manuscript before final submission.

Reviewer #3 (Remarks to the Author):

In this paper, the authors present a new methodology for creating hybrid grafted perfect vector vortex beams (GPVVBs) with spatial variant polarization profiles. The features that were pointed out have been well-revised. The authors provided a detailed explanation of the limits of superposing different GPVVBs on a single metasurface, as demonstrated through simulations and experiments on a 400 nm by 400 nm metasurface. The metasurface exhibits four higher-order GPVVBs in both simulation and experiments (added as Fig. S7, Fig. S11) which further increases the information in the light beams. In addition, by controlling the axicon period, the authors mitigate the problem of crosstalk between rings caused by parasitic light, a conventionally observed problem with perfect vortices. The conversion efficiency of the proposed Au nanorod was additionally illustrated in Fig. S4 and Fig. S5, with the potential for further enhancement by using dielectric materials. Overall, the manuscript has been significantly improved, addressing all previously raised comments, so I would recommend publication in Nature Communications.

Response to reviewers' comments

We thank all the reviewers for their positive comments.

Reviewer #1:

The authors have considered the remarks. I believe the content is now clear and sufficient to be reproduced by others. Following on my previous comment, I still believe that an optical system operating with less than 10% of efficiency would have hard time to find realistic application. In the future, maybe, some solution would be needed to make this approach useful.

Reply: We thank the reviewer for the positive comment.

To clarify this point on low efficiency issue, we added one sentence in the discussion “The demonstrated metasurfaces consisting of gold nanorods have a low conversion efficiency, which can be dramatically increased with dielectric metasurfaces (e.g., titanium dioxide)³⁷.” (Line 279-281, Page 10)

We also added a reference on dielectric metasurfaces with titanium dioxide.

37. Jin L, et al. Dielectric multi-momentum meta-transformer in the visible. *Nat. Commun.* 10, 4789 (2019).

Reviewer #2:

I would like to thank the authors for their detailed responses and the changes made to the manuscript and supplementary material. Please consider highlighting the possible use of TiO₂ nano pillars for this application within the manuscript - this would also encourage other researchers to explore this opportunity. Lastly, please thoroughly proofread the manuscript before final submission.

Reply: We thank the reviewer for the positive comment and valuable suggestion.

To clarify this point on low efficiency issue, we added one sentence in the discussion “The demonstrated metasurfaces consisting of gold nanorods have a low conversion efficiency, which can be dramatically increased with dielectric metasurfaces (e.g., titanium dioxide)³⁷.” (Line 279-281, Page 10)

We also added a reference on dielectric metasurfaces with titanium dioxide.

37. Jin L, et al. Dielectric multi-momentum meta-transformer in the visible. *Nat. Commun.* 10, 4789 (2019).

Reviewer #3:

In this paper, the authors present a new methodology for creating hybrid grafted perfect vector vortex beams (GPVVBs) with spatial variant polarization profiles. The features that were pointed out have been well-revised. The authors provided a detailed explanation of the limits of superposing different GPVVBs on a single metasurface, as demonstrated through simulations and experiments on a 400um by 400 um metasurface. The metasurface exhibits four higher-order GPVVBs in both simulation and experiments (added as Fig. S7, Fig. S11) which further increases the information in the light beams. In addition, by controlling the axicon period, the authors mitigate the problem of crosstalk between rings caused by parasitic light, a conventionally observed problem with perfect vortices. The conversion efficiency of the proposed Au nanorod was additionally illustrated in Fig. S4 and Fig. S5, with the potential for further enhancement by using dielectric materials. Overall, the manuscript has been

significantly improved, addressing all previously raised comments, so I would recommend publication in Nature Communications.

Reply: We thank the reviewer for the very positive comment.